# LSTMs and Deep Residual Networks for Carbohydrate and Bolus Recommendations in Type 1 Diabetes Management

**DOI:** 10.3390/s21093303

**Published:** 2021-05-10

**Authors:** Jeremy Beauchamp, Razvan Bunescu, Cindy Marling, Zhongen Li, Chang Liu

**Affiliations:** 1School of Electrical Engineering and Computer Science, Ohio University, Athens, OH 45701, USA; jb199113@ohio.edu (J.B.); marling@ohio.edu (C.M.); zl967815@ohio.edu (Z.L.); liuc@ohio.edu (C.L.); 2Department of Computer Science, University of North Carolina at Charlotte, Charlotte, NC 28223, USA

**Keywords:** diabetes management, deep learning, artificial intelligence

## Abstract

To avoid serious diabetic complications, people with type 1 diabetes must keep their blood glucose levels (BGLs) as close to normal as possible. Insulin dosages and carbohydrate consumption are important considerations in managing BGLs. Since the 1960s, models have been developed to forecast blood glucose levels based on the history of BGLs, insulin dosages, carbohydrate intake, and other physiological and lifestyle factors. Such predictions can be used to alert people of impending unsafe BGLs or to control insulin flow in an artificial pancreas. In past work, we have introduced an LSTM-based approach to blood glucose level prediction aimed at “what-if” scenarios, in which people could enter foods they might eat or insulin amounts they might take and then see the effect on future BGLs. In this work, we invert the “what-if” scenario and introduce a similar architecture based on chaining two LSTMs that can be trained to make either insulin or carbohydrate recommendations aimed at reaching a desired BG level in the future. Leveraging a recent state-of-the-art model for time series forecasting, we then derive a novel architecture for the same recommendation task, in which the two LSTM chain is used as a repeating block inside a deep residual architecture. Experimental evaluations using real patient data from the OhioT1DM dataset show that the new integrated architecture compares favorably with the previous LSTM-based approach, substantially outperforming the baselines. The promising results suggest that this novel approach could potentially be of practical use to people with type 1 diabetes for self-management of BGLs.

## 1. Introduction and Motivation

Diabetes self-management is a time-consuming, yet critical, task for people with type 1 diabetes. To avoid serious diabetic complications, these individuals must continually manage their blood glucose levels (BGLs), keeping them as close to normal as possible. They must avoid both low BGLs, or hypoglycemia, and high BGLs, or hyperglycemia, for their physical safety and well-being. Diabetes self-management entails carefully monitoring BGLs throughout the day, by testing blood obtained from finger sticks and/or using a continuous glucose monitoring (CGM) system. It also entails making numerous daily decisions about the timing and dosage of insulin and the timing, ingredients, and quantity of food consumed.

Current diabetes self-management may be characterized as reactive, rather than proactive. When BGLs are too high, individuals may take insulin to lower them, and when BGLs are too low, they may eat a snack or take glucose tablets to raise them. The ability to accurately predict BGLs could enable people with type 1 diabetes to take preemptive actions before experiencing the negative effects of hypoglycemia or hyperglycemia. There have been efforts to model BGLs for the purpose of determining insulin dosages dating back to the 1960s [1]. There has been much recent work in BGL prediction for the purpose of providing support for diabetes self-management, including our own [2,3]. Accounts of some of the most recent BGL prediction efforts can be found in the proceedings of two international BGL prediction challenges [4,5]. It should be noted that even with the benefit of accurate BGL predictions, individuals still need to determine how much to eat, how much insulin to take, and what other actions they can take to prevent hypoglycemia or hyperglycemia.

This research aims to essentially reverse the BGL prediction problem, instead predicting the amount of carbohydrate (carbs) an individual should consume or the amount of insulin they should take to reach a BGL target. Previously, we presented a new LSTM-based neural architecture trained to answer what-if questions, such as “What will my BGL be in 60 min if I have a meal containing 50 carbs in 10 min?” [6]. In subsequent work [7], we showed that, using the BGL target as a feature and carbs or insulin as the label, a similar LSTM-based architecture could be trained to predict how much carbohydrate to eat or how much insulin to take during the prediction window to achieve that BGL target. Preliminary results were reported in [7] only for the task of carbohydrate recommendation, where the aim was to achieve a desired target BGL 30 or 60 min into the future. The timing of the meal was variable within the prediction window and was used as one of the inputs to the model. In this paper, we update the task definition to make it more applicable to the type of situations that are frequently encountered in the self-management of type 1 diabetes. As such, the timing of the bolus or meal is now fixed at 10 min into the future, based on the assumption that patients are most interested in using the system right before making a meal or bolus decision. To achieve the desired BGL, the user can specify any time horizon between 30 and 90 min, giving them more flexibility in terms of how fast they want their BGL to change. Furthermore, we improve the LSTM-based architecture from [7] and use it as a repeating residual block in a deep residual forecasting network derived from the BGL prediction architecture recently proposed by Rubin-Falcone et al. [8]. The neural architecture from [8] is in turn a modified version of the N-BEATS architecture [9] that was shown to obtain state-of-the-art results on a wide array of time series forecasting problems. Overall, the new recommendation approach using the integrated deep network is generic in the sense that it can be trained to make recommendations about any variable that can impact BGLs, in particular, carbohydrates and insulin. Carbohydrate recommendations are potentially useful when someone wants to prevent hypoglycemia well in advance or when someone wants to achieve a higher target BGL before physical exercise that is expected to lower it. Bolus recommendations are useful prior to meals and for lowering BGLs when individuals experience hyperglycemia.

The rest of the paper is organized as follows: Section 2 presents related work in blood glucose level prediction and automated bolus calculators, and positions our work in relation to recent developments in these areas. Three major recommendation scenarios are introduced in Section 3, followed in Section 4 by a description of a set of baseline models and neural architectures that are designed to make recommendations in each of these scenarios. Section 5 introduces the OhioT1DM dataset and explains how recommendation examples were derived from it. The experimental methodology and results are presented in Section 6 and Section 7, respectively. The paper concludes in Section 8 with a summary of our contributions and ideas for future work.

## 2. Related Work

Bolus calculators, which recommend insulin dosages using standard formulas, have been in use since 2003 [10]. They typically base their recommendations on carbohydrate intake, carbohydrate-to-insulin ratio, insulin on board, and target BGL. As described in Walsh et al. [11], current bolus calculators are error prone, but they have room for potential improvement. One such improvement mentioned was the use of the massive amount of clinical data that is collected from these bolus advisor systems. AI techniques have been used to take advantage of this data to create more intelligent and personalized insulin recommendation systems. Pesl et al. [12] describe a bolus advisor system based on case-based reasoning that personalizes an individual’s bolus calculations. The system gathers simple information, such as the range of recent blood glucose levels and the time of day, and compares the current situation to situations from the past to find similar cases. The system then uses the bolus recommendation from a similar previous case and adapts it to the current scenario. The work by Tyler et al. [13] shows a K-nearest-neighbors-based system that provides weekly recommendations to improve the effectiveness of an individual’s multiple daily injection therapy. With the amount of clinical data collected from CGM systems and wearable sensors, deep learning is a natural fit for insulin advisor systems. The work in Mougiakakou and Nikita [14] represents an early attempt at creating insulin recommendations by using neural networks. Cappon et al. [15] observe that the standard formula approach to bolus calculation ignores potentially important preprandial conditions, such as the glucose rate of change. To address this, they propose a simple feed-forward neural network that uses CGM data and other easily accessible factors to personalize the bolus calculation. They demonstrate a small, but statistically significant, improvement in the blood glucose risk index, using simulated data for experimental evaluation. Sun et al. [16] also use simulated data to train a basal-bolus advisor using reinforcement learning. Their system aims to provide personalized suggestions to people with type 1 diabetes taking multiple daily injections of insulin. Zhu et al. [17] also use simulated type 1 diabetes data for their deep reinforcement learning approach to personalizing bolus calculations. They use the UVA/Padova type 1 diabetes simulator [18] to train models to learn how to reduce or amplify the bolus dose recommended by a bolus calculator to provide more personalized recommendations.

In contrast with previous work that used a type 1 diabetes simulator [15,16,17], the systems described in this paper are trained and evaluated on data acquired from people with type 1 diabetes, derived from the OhioT1DM dataset [19] as explained in Section 5. The case-based reasoning system introduced by Pesl et al. [12] also makes use of real patient data; however, their system does not learn directly from this data. Instead, it learns from clinical experts’ advice, requiring also that the system be tweaked on a regular basis. Elsewhere [6], we have shown that patterns that are learned from simulated data do not always transfer to real data, and vice-versa. By training and evaluating on real data, the results reported in this paper are expected to be more representative of the actual performance of the recommendation system if it were to be deployed in practice. Most of the related work on bolus recommendations presented above use global means of evaluating system performance, such as the percentage of time BGLs are in a target range [16,17], or the blood glucose risk index [15]. In contrast, our approaches are evaluated retrospectively on how close their recommendations are to the actual carbohydrate content or bolus dosages that led to a particular target BGL. As such, the trained models can be used to make recommendations to achieve a specific BGL. The neural architectures that we propose in this paper are also general in the sense that they can be used to make recommendations for any type of discrete intervention that may impact BGLs. Although, in this paper, they are evaluated on bolus, carbs, and bolus given carbs recommendations, we also see them as applicable for recommending other relevant variables, such as exercise.

## 3. Three Recommendation Scenarios

The system relies on the following input data: (1) BGLs, measured at 5-minute intervals through a CGM system; (2) discrete deliveries of insulin (boluses) and continuous infusions of insulin (basal rates), recorded by an insulin pump; and (3) mealtimes and carbohydrate estimates, self-reported by subjects. Given the available data up to and including the present (time *t*), the system aims to estimate how much a person should eat or bolus 10 min from now (time t+10) such that their blood glucose will reach a target level τ minutes after that action (time t+10+τ). A system that computes these estimates could then be used in the following three recommendation scenarios:Carbohydrate Recommendations Estimate the amount of carbohydrate $C_{t+10}$ to have in a meal to achieve a target BG value $G_{t+10+\tau }$.Bolus Recommendations: Estimate the amount of insulin $B_{t+10}$ to deliver with a bolus to achieve a target BG value $G_{t+10+\tau }$.Bolus Recommendations given Carbohydrates: Expecting that a meal with $C_{t+20}$ grams of carbohydrate will be consumed 20 min from now, estimate the amount of insulin $B_{t+10}$ to deliver with a bolus 10 min before the meal to achieve a target BG value $G_{t+10+\tau }$. We used the 10-minute interval based on the recommendation given by the physician to the subjects, which was to bolus 10 min before the meal.

These recommendation scenarios were designed to align with decision-making situations commonly encountered by people with type 1 diabetes. In particular, the corresponding recommendation systems would help an individual to estimate how much to eat or bolus for the purpose of raising or lowering their BGL (scenarios 1 and 2), as well as how much to bolus for a planned meal (scenario 3).

In the following Section 4, we describe several baseline models and neural architectures, all implementing the three types of recommendations. The neural architectures use Long Short-Term Memory (LSTM) networks either in a standalone prediction model (Section 4.1) or integrated as basic repeating blocks in a deep residual network (Section 4.2). The models are trained on examples extracted from the OhioT1DM dataset [19], as explained in Section 5. Ideally, to match the intended use of these recommendations in practice, training examples should not have any extra meals or boluses in the prediction window [t,t+10+τ]. Following the terminology from [6], we call these examples inertial. However, to benefit from a larger number of training examples, we also train and evaluate models on a more general class of unrestricted examples, in which other bolus or meal events can appear in the prediction window. Correspondingly, experimental results for inertial vs. unrestricted examples are presented in Section 7.

## 4. Baseline Models and Neural Architectures

Given training data containing time series of blood glucose levels, meals with their carbohydrate intake, and boluses with their corresponding insulin dosages, we define the following two baselines:Global averageFor the carbohydrate recommendation scenario, the average number μ of carbs over all the meals in the subject’s training data are computed and used as the estimate for all future predictions for that subject, irrespective of the context of the example. Analogously, for the bolus and bolus given carbs recommendation scenarios, μ is the average amount of insulin dosage over all boluses in the subject’s training data. This is a fairly simple baseline, as it predicts the same average value for every test example for a particular subject.ToD average: In this Time-of-Day (ToD)-dependent baseline, an average number of carbs or an average amount of bolus insulin is computed for each of the following five time windows during a day:
12 am–6 am: $\mu_{1}$ = early breakfast/late snacks.6 am–10 am: $\mu_{2}$ = breakfast.10 am–2 pm: $\mu_{3}$ = lunch.2 pm–6 pm: $\mu_{4}$ = dinner.6 pm–12 am: $\mu_{5}$ = late dinner/post-dinner snacks.For each ToD interval, the corresponding average is computed over all the meal events or boluses appearing during that interval in the subject’s training data. At test time, we determine the ToD interval overlapping t+10 and output the associated ToD average.

Although the ToD average baseline may perform well for subjects who have regular eating patterns, e.g., eating the same number of carbs for breakfast at the same time each morning, the baseline is expected to do poorly for subjects whose eating schedules and carbohydrate amounts are more variable.

### 4.1. LSTM Models for Carbohydrate and Insulin Recommendation

The history of previous BGL values, insulin (boluses and basal rates), and meals can significantly alter the impact that a meal or bolus will have on future BG levels. Since this information is ignored by the two baselines, their performance is likely to be suboptimal. To use all relevant time series data, we propose the LSTM-based architectures shown in Figure 1 for carbohydrate recommendation and Figure 2 for bolus recommendation. The first Long Short-Term Memory (LSTM) network [20], LSTM${}_{1}$, is unrolled over the previous 6 h of data, up to and including the present time *t*. At every 5-min time step, the LSTM${}_{1}$ network takes as input the BG level, the carbohydrate amounts, and the insulin dosages (if any) recorded for that time step. Although LSTM${}_{1}$ is sufficient for processing the input of inertial examples, it cannot be used to process events that appear in the prediction window (t,t+10+τ) of unrestricted examples, for which BGL values are not available. Therefore, when training on unrestricted examples, we also use a second LSTM model, LSTM${}_{2}$, whose initial state is computed by projecting the final state of LSTM${}_{1}$ at time *t* using a linear transformation. The LSTM${}_{2}$ network is then run over all time steps in the prediction window (t,t+10+τ). The final states of LSTM${}_{1}$ and LSTM${}_{2}$ are concatenated and fed as input to a fully connected network (FCN) that outputs an estimate of the carbs or bolus insulin at time step t+10. In addition to the LSTM final state(s), the FCN also uses the following features as input:The target blood glucose level τ+10 min into the future, i.e., $G_{t+10+\tau }$.The prediction horizon τ.The ToD average for the time frame that contains t+10.For the bolus given carbs scenario only, the planned amount $C_{t+20}$ of carbohydrate becomes part of the input, too.

Each LSTM uses vectors of size 32 for the states and gates, whereas the FCN is built with up to 5 hidden layers, each consisting of 64 ReLU neurons, and one linear output node. Please note that by using the final state of LSTM${}_{1}$ to initialize LSTM${}_{2}$, the latter’s final state should theoretically be able to capture any useful information that is represented in the final state of LSTM${}_{1}$, which may call into question the utility of concatenating the two final states. This architectural decision is nevertheless supported empirically through evaluations on the validation data, which show improvements in prediction performance when both states are used (Section 6.3).

### 4.2. Deep Residual Models for Carbohydrate and Insulin Recommendation

Oreshkin et al. [9] have recently introduced a new architecture for time series forecasting, the Neural Basis Expansion for Interpretable Time-Series Forecasting (N-BEATS). The basic building block of N-BEATS is a fully connected structure that initially takes as input a fixed-size lookback period of past values of the target variable and outputs both forecast (estimates of future values) and backcast (estimates of past values) vectors. Blocks are organized into stacks such that the backcast of the current block is subtracted from its input and fed as input to the next block, whereas the forecast vectors from each block are summed up to provide the overall stack forecast. The stacks themselves are chained in a pipeline where the backcast output of one stack is used as input for the next stack. The overall model forecast is then computed by accumulating the forecasts across all the stacks.

The N-BEATS architecture was shown in [9] to obtain state-of-the-art performance on a wide range of time series prediction tasks, which suggests that it can serve as a model of choice for BGL prediction, too. However, in BGL prediction, time series of variables other then the primary blood glucose are also available. Rubin-Falcone et al. [8] adapted the N-BEATS architecture to allow secondary, sparse variables such as meals and bolus insulin to be used as input, whereas backcasting was still performed only on the primary forecasting variable, blood glucose. Furthermore, the fully connected structure of the basic N-BEATS block was replaced with LSTMs to better model the temporal nature of the input. The last LSTM state was used as input to one fully connected layer whose output was split into the backcast and forecast vectors. Additional per-block forecast and backcast loss terms were also added to provide more supervision.

We adapted the deep residual network from [8] to perform carb or bolus recommendations using the LSTM-based architecture from Section 4.1 to instantiate each block in the stack, as shown in Figure 3. Compared to the architecture from [8], the most significant differences are:The use of a chain of two LSTM networks in each block.The inclusion of additional inputs to the fully connected layers, i.e., the target BG level, the time horizon, and the ToD average.Although backcasting is still done for blood glucose, forecasting is done for carbs or bolus, depending on the recommendation scenario.

Although Oreshkin et al. [9] used 30 blocks and Rubin-Falcone et al. [8] used 10 blocks, the validation experiments for the recommendation tasks showed that the most effective deep residual architecture uses only up to 5 blocks, depending on the recommendation scenario (Section 6.3).

## 5. Using the OhioT1DM Dataset for Recommendation Examples

To evaluate the proposed recommendation models, we create training and test examples based on data collected from 12 subjects with type 1 diabetes that is distributed with the OhioT1DM dataset [19]. The 12 subjects are partitioned in two subsets as follows:OhioT1DM 2018: This is the first part of the dataset, containing data collected from 6 patients. It was used for the 2018 Blood Glucose Level Prediction (BGLP) challenge [4].OhioT1DM 2020: This is the second part of the dataset, containing data collected from 6 additional patients. It was used for the 2020 BGLP challenge [5].

Time series containing the basal rate of insulin, boluses, meals, and BGL readings were collected over 8 weeks, although the exact number of days varies from subject to subject. Insulin and BGL data were automatically recorded by each subject’s insulin pump. Meal data were collected in two different ways. Subjects self-reported mealtimes and estimated carbs via a smartphone interface. Subjects also entered estimated carbs into a bolus calculator when bolusing for meals, and these data were recorded by the insulin pump.

### 5.1. The Bolus Wizard

To determine their insulin dosages, the subjects in the OhioT1DM study used a bolus calculator, or ”Bolus Wizard (BW),” which was integrated in their insulin pumps. They used it to calculate the bolus amount before each meal as well as when using a bolus to correct for hyperglycemia. To use the BW, a subject enters their current blood glucose level and, if eating, their estimated number of grams of carbohydrate. To calculate a recommended insulin dosage, the BW uses this input from the subject, plus the amount of active insulin the subject already has in their system, along with the following three pre-programmed, patient-specific parameters:The carb ratio, which indicates the number of grams of carbohydrate that are covered by a unit of insulin.The insulin sensitivity, which tells how much a unit of insulin is expected to lower the subject’s blood glucose level.The target blood glucose range, which defines an individual’s lower and upper boundaries for optimal blood glucose control.

All three parameters may vary, for the same individual, throughout the day and over time (https://www.medtronicdiabetes.com/loop-blog/4-features-of-the-bolus-wizard, accessed on 25 April 2021). Given this input and these parameters, the BW calculates the amount of insulin the subject should take to maintain or achieve a blood glucose level within their target range. The calculation is displayed to the subject as a recommendation, which the subject may then accept or override.

Based on the inputs and the patient-specific parameters described above, the BW uses a deterministic formula to calculate the bolus amount before each meal. As such, when trained in the bolus given carbs recommendation scenario, there is the risk that the deep learning models introduced in Section 4 might simply learn to reproduce this deterministic dependency between bolus and carbs, while ignoring the target BG level that is used as input. However, this is not the case in our experimental settings, for the following reasons:The machine learning (ML) models do not have access to any of the three patient-specific parameters above, which can change throughout the day and over time, and which are set based on advice from a health care professional.The BW uses a fixed target BG range depending on the time of day, whereas the target in the recommendation scenarios is a more specific BG level, to be achieved at a specific time in the near future.The amount of insulin calculated by the BW is only a recommendation, which is often overridden by subjects. We ran an analysis of the OhioT1DM dataset in which we counted how many times the amount of insulin that was actually delivered was different from the bolus recommendation. The analysis revealed that of all the times that the BW was used, its recommendation was overridden for about a fifth of the boluses. Furthermore, there are subjects in the dataset who often did not use the BW (540 and 567), or who chose to not use the BW at all (596).

Therefore, the ML models will have to go beyond using solely the carbohydrate amount in the intended meal. To fit the bolus recommendation examples, they will need to learn the impact that a bolus has on the target BG level for the specified prediction horizon, taking into account the amount of carbohydrate in the meal as well as the history of carbs, insulin, and BG levels. This data driven approach to bolus recommendation relieves the physician from the cognitively demanding task of regularly updating parameters such as the carb ratio and the insulin sensitivity, which often requires multiple fine-tuning steps. In contrast, any relevant signal that is conveyed through the carb ratio and insulin sensitivity is expected to be learned by the ML models from the data.

### 5.2. Pre-Processing of Meals and BG Levels

While exploring the data, it was observed that self-reported meals and their associated boluses were in unexpected temporal positions relative to each other. For many meals, patients recorded a timestamp in the smartphone interface that preceded the corresponding bolus timestamp recorded in the insulin pump. This was contrary to what was recommended to the subjects by their physician, which was to bolus 10 min before the meal. This discrepancy is likely due to subjects reporting incorrect mealtimes in the smartphone interface.

To correct the meal events, we used the data input to the BW in the insulin pump and ran a pre-processing step that changed the timestamp of each meal associated with a bolus to be exactly 10 min after that bolus. For these meals, we also used the number of carbs provided to the BW, which is likely to be more accurate than the estimate provided by the subject through the smartphone interface. To determine the self-reported meal event associated with a bolus with non-zero carb input, we searched for the meal that was closest in time to the bolus, within one hour before or after it. In case there were two meals that are equally close to the bolus, we selected the one for which the number of carbs from the smartphone interface was closest to the number of carbs entered into the BW. If no self-reported meal was found within one hour of the bolus, it was assumed that the subject forgot to log their meal on the smartphone interface. As such, a meal was added 10 min after the bolus, using the amount of carbs specified in the BW for that bolus. Ablation results reported in Section 6.2 show that this pre-processing of meal events leads to significantly more accurate predictions, which further justifies the pre-processing.

All gaps in BGL data are filled in with linearly interpolated values. However, we filter out examples that meet any of the following criteria:The BGL target is interpolated.The BGL at present time *t* is interpolated.There are more than 2 interpolated BGL measurements in the one hour of data prior to time *t*.There are more than 12 interpolated BGL measurements in the 6 h of data prior to time *t*.

### 5.3. Mapping Prototypical Recommendation Scenarios to Datasets

According to the definition given in Section 3, the carbohydrate recommendation scenario refers to estimating the amount of carbohydrate $C_{t+10}$ to have in a meal to achieve a target BG value $G_{t+10+\tau }$. This is done using the history of data up to and including the present time *t*. However, many carbohydrate intake events $C_{t+10}$ are regular meals, which means that they are preceded by a bolus event at time *t*. Since in the carbohydrate recommendation scenario we are especially interested in scenarios where the subject eats to correct or prevent hypoglycemia, we created two separate datasets for carbohydrate prediction:Carbs${}^{\left(\pm b\right)}$: this will contain examples for all carbohydrate intake events, with (+b) or without (−b) an associated bolus.Carbs${}^{\left(-b\right)}$: this will contain examples only for carbohydrate intake events without (−b) an associated bolus.

Most of the Carbs${}^{\left(-b\right)}$ examples are expected to happen in one of three scenarios: (1) when correcting for hypoglycemia; (2) before exercising; and (3) when having a bedtime snack to prevent nocturnal hypoglycemia. Given that they are only a small portion of the overall carbohydrate events, in Section 7 we present results for both Carbs${}^{\left(\pm b\right)}$ and Carbs${}^{\left(-b\right)}$ recommendation scenarios.

Furthermore, mirroring the two bolus recommendation scenarios introduced in Section 3, we introduce the following notation for the corresponding datasets:Bolus${}^{\left(\pm c\right)}$: this will contain examples for all bolus events, with (+c) or without (−c) an associated carbohydrate intake.Bolus${}^{\left(+c\right)}$: this will contain examples only for the bolus events with (+c) an associated carbohydrate intake.

The three major recommendation scenarios introduced in Section 3 can then be mapped to the corresponding datasets as follows:Carbohydrate Recommendations: Estimate the amount of carbohydrate $C_{t+10}$ to have in a meal to achieve a target BG value $G_{t+10+\tau }$.
Carbs${}^{\left(-b\right)}$, inertial: this reflects the prototypical scenario where a carbohydrate intake is recommended to correct or prevent hypoglycemia.Bolus Recommendations: Estimate the amount of insulin $B_{t+10}$ to deliver with a bolus to achieve a target BG value $G_{t+10+\tau }$.
Bolus${}^{\left(\pm c\right)}$, inertial: this reflects the prototypical scenario where a bolus is recommended to correct or prevent hyperglycemia. Because in the inertial case a carb event cannot appear after the bolus, this could also be denoted as Bolus${}^{\left(-c\right)}$.Bolus Recommendations given Carbohydrates: Expecting that a meal with $C_{t+20}$ grams of carbohydrate will be consumed 20 min from now, estimate the amount of insulin $B_{t+10}$ to deliver with a bolus 10 min before the meal to achieve a target BG value $G_{t+10+\tau }$.
Bolus${}^{\left(+c\right)}$, inertial: this reflects the prototypical scenario where a bolus is recommended before a meal.

### 5.4. Carbohydrate and Bolus Statistics

Table 1 shows the number of carbohydrate events in each subject’s pre-processed data, together with the minimum, maximum, median, average, and standard deviation for the number of carbs per meal. Overall, the average number of carbs per meal is between 22 and 69, except for subjects 570 and 544 whose meal averages and standard deviations are significantly larger. Table 2 shows similar statistics for boluses and their dosages, expressed in units of insulin. Overall, the number of boluses is more variable than the number of meals. There is also a fairly wide range of average bolus values in the data, with subject 567 having a much higher average than other subjects. It is also interesting to note that subject 570, who had the largest average carbs per meal, had more than twice the number of boluses than any other subject while at the same time having the lowest average bolus. Subject 570 also used many dual boluses, which we did not use as prediction labels because the scope of the project covers only recommendations for regular boluses.

### 5.5. From Meals and Bolus Events to Recommendation Examples

In all recommendation scenarios, the prediction window ranges between the present time *t* and the prediction horizon t+10+τ. For the carbohydrate or bolus recommendation scenarios, the meal or the bolus is assumed to occur at time t+10. For the bolus given carbs scenario, the bolus occurs at time t+10 and is followed by a meal at time t+20, which matches the pre-processing of the meal data. For evaluation purposes, we set τ to values between 30 and 90 min with a step of 5 min, i.e., τ∈{30,35,40,…,90} for a total of 13 different values. As such, each meal/bolus event in the data results in 13 recommendation examples, one example for each value of τ. Although all 13 examples use the same value for the prediction label, e.g., $B_{t+10}$ for bolus prediction, they will differ in terms of the target BG feature $G_{t+10+\tau }$ and the τ feature, both used directly as input to the FC layers in the architectures shown in Figure 1 and Figure 2. For the bolus given carbs scenario, the 13 examples are only created when there is a meal that had a bolus delivered 10 min prior. Due to the way the data are pre-processed, it is guaranteed that if a meal had a bolus associated with it, the bolus will be exactly 10 min before the meal.

The fast-acting insulin that the subjects use has its strongest glucose lowering effect 1 to 1.5 h after it is taken (https://www.medtronicdiabetes.com/sites/default/files/library/download-library/workbooks/BasicsofInsulinPumpTherapy.pdf, accessed 25 April 2021). We decided to limit the prediction horizon to this 1.5-h limit for two main reasons. First, extending it beyond 90 min in the inertial scenario would lead to significantly fewer examples for those time horizons, because the inertial scenario requires that no other events happen during the prediction window. Second, the longer the prediction window, the more likely that hidden variables occur that have an impact on the BGL, and thus cause the model to underperform. We use the term hidden variables to refer to any relevant subject-specific factors or physical activities such as walking or exercise that are either not reported or under-reported in the data.

Table 3 shows the number of *inertial* examples for 5 prediction horizons, as well as the total over all 13 possible prediction horizons. Table 4 shows the number of *unrestricted* examples. Since the same number of unrestricted examples are available for every prediction horizon, only the totals are shown. The only exceptions would be if an event were near the end of a subject’s data and the prediction horizon t+10+τ goes past the end of the dataset for some value of τ.

For the carbohydrate and bolus given carbs recommendation scenarios, the gap between the number of *inertial* and *unrestricted* examples is not very large, as most examples qualify as inertial examples. However, in the bolus recommendation scenario, there is a very sizable gap between the number of inertial vs. unrestricted examples. This is because a significant number of boluses are associated with meals, and since these meals are timestamped to be 10 min after the bolus, the result is that a bolus at time t+10 will be associated with a meal at time t+20. Therefore, for preprandial boluses at t+10, the meal at time t+20 will prohibit the creation of inertial recommendation examples, because by definition inertial examples do not allow the presence of other events in the prediction window (t,t+10+τ).

## 6. Experimental Methodology

For each of the 12 subjects in the dataset, their time series data are split into three sets, as follows:Testing: the last 10 days of data.Validation: the 10 days of data preceding the testing portion.Training: the remainder of the data, around 30 days.

The blood glucose, carbs, and insulin values are all scaled to be between [0,1] using maximum and minimum values computed over training data. When computing the performance metrics at test time, the predicted values are scaled back to the original range. The training loss function is set to be the mean squared error between the meal or bolus values recorded in the data and the estimates computed by the output node of the fully connected layers in the LSTM-based architectures, or by the accumulated forecasts in the N-BEATS models. The Adam [21] gradient-based algorithm is used for minimizing the loss during training, for which the learning rate and the mini-batch size are tuned on the validation data. To alleviate overfitting, dropout and early stopping with an inertia of 10 epochs are used in all experiments.

Before training a personalized model for a specific subject, a generic model is first pre-trained on the union of all 12 subjects’ training data. The generic model is then fine-tuned separately for each individual subject, by continuing training on that subject’s training data only. The pre-training allows the model parameters to be in a better starting position before fine-tuning, allowing faster and better training. The learning rate and batch size are tuned for each subject on their validation data. For each subject, the results are aggregated over 10 models that are trained with different seedings of the random number generators.

The metrics used to evaluate the models are the Root Mean Squared Error (RMSE) and the Mean Absolute Error (MAE). Two scores are reported for each of the LSTM-based and N-BEATS-based recommendation models:The 〈**model**〉**.mean** score calculates the average RMSE and MAE on the testing data across the 10 models trained for each subject, and then averages these scores across all subjects.The 〈**model**〉**.best** score instead selects for each subject the model that performed best in terms of MAE on the validation data, out of the 10 models trained for that subject. The RMSE and MAE test scores are averaged over all subjects.

The units for RMSE and MAE are grams (g) for carbohydrate recommendations and units of insulin (u) for bolus recommendations. Two sets of models were trained for each recommendation scenario: a set of models was trained and evaluated on *inertial* examples and a set was trained and evaluated on *unrestricted* examples.

### 6.1. Subject Selection for Testing in Each Recommendation Scenario

Although using both the 2018 and 2020 subsets of the OhioT1DM Dataset [19,22] provides us with data from 12 total subjects, not all 12 can be used in each scenario, due to insufficient examples in their respective development or test subsets. The subjects whose data were used or not at test time are listed below for each scenario, together with a justification:*Carbs${}^{\left(\pm b\right)}$ Recommendation*: Subjects 567 and 570 were left out at test time. Subject 567 had 0 meal events in the testing portion of their data. Subject 570 frequently used dual boluses; as such, there were very few inertial examples for this subject at all. Of the few inertial examples that were available, 0 were in the testing or validation portions of the data.*Carbs${}^{\left(-b\right)}$ Recommendation*: Due to the limited number of examples for this scenario, we trained and evaluated models only for the subjects whose data contained at least 50 carb events with no associated bolus. These are subjects 559, 575, 588, and 591. Although subject 596 also had a sufficient number of carb events, we discovered that all carbohydrate inputs for their BW were 0. As a consequence of this missing data, it cannot be determined which boluses were used for BGL correction, and which were used to cover meals. Therefore, subject 596 cannot be used in this scenario.*Bolus${}^{\left(\pm c\right)}$ Recommendation*: Subjects 544 and 567 were left out at test time. Subject 544 had few inertial examples overall, and 0 in the validation portion of the data. This is because most bolus events in their data were used in conjunction with a meal. Similar to the carbohydrate recommendation scenario, subject 567 was not used in this scenario because of the lack of meal events in their test data. The missing meal data would make the bolus recommendation results for this subject unrealistic and indistinguishable between the inertial and unrestricted cases.*Bolus${}^{\left(+c\right)}$ Recommendation*: Subjects 567, 570, and 596 were left out at test time. As explained for other scenarios above, subject 567 had 0 meals in the test portion of their data. For subject 570, there were 0 inertial examples in the test portion. As explained for the Carbs${}^{-b}$ recommendation scenario, due to missing BW data, for subject 596 it cannot be determined which boluses were used for BGL correction, and which were used to cover meals, so their data cannot be used in this scenario, either.

Irrespective of which subjects are used at test time, the data from all 12 patients is used for pre-training purposes in each recommendation scenario. Furthermore, the set of subjects stays consistent between the inertial and unrestricted cases for any given recommendation scenario.

### 6.2. Evaluating the Impact of Pre-Processing of Meals

To determine the utility of the pre-processing of meals procedure introduced in Section 5.2, we trained and evaluated N-BEATS-based models for the carbohydrate recommendation scenario Carbs${}^{\left(\pm b\right)}$ using the original data vs. using the pre-processed data. When training on pre-processed data, we report in Table 5 two validation results: when evaluating on all the pre-processed meals in the validation data (pre${}^{+}$) vs. evaluating only on meals that were not added during pre-processing (pre${}^{-}$). The results show that in both cases the pre-processing of meals leads to statistically significant improvements in RMSE and MAE. Pre-processing of meals also benefits the bolus recommendation scenario, as shown in Table 6. These results can be seen as further evidence of the fact that the meal timestamps recorded in the smartphone interface are unreliable and that mealtimes should instead be anchored to the bolus timestamps recorded by the BW, as done in the pre-processing procedure.

### 6.3. Tuning the Architecture and the Hyper-Parameters on the Development Data

Table 7 show the results of the LSTM- and N-BEATS-based models, with vs. without using the final state produced by the LSTM${}_{1}$ component as input to the fully connected network. The results show that using the final state from LSTM${}_{1}$ directly as input leads to a substantial improvement for the carbohydrate recommendation scenario Carbs${}^{\left(\pm b\right)}$, while maintaining a comparable performance for the bolus recommendation scenario. Consequently, in all remaining experiments the architecture is set to use the final state of LSTM${}_{1}$ as input to the FC layers.

In the original N-BEATS model of Oreshkin et al. [9], the backcast and forecast outputs of each block are produced as the result of two separate fully connected layers. In the block architecture shown in Figure 1, Figure 2 and Figure 3 however, the *FC Layers* component uses just one final fully connected layer to produce both backcast and forecast values. The results in Table 8 show that overall, using a joint final layer is competitive or better than using separate layers.

For each prediction scenario, the hyper-parameters for both the LSTM-based and N-BEATS-based models were tuned on development data. The inertial and unrestricted models are tuned independent of each other. The learning rate was tuned by monitoring the learning curves, using values between 0.0002 [8] and 0.1. After multiple experiments, a fixed learning rate of 0.001 was observed to give the best results on development data in all scenarios. The number of blocks in N-BEATS, the number of FC layers in the LSTM, and the dropout rate were then tuned in that order. The number of N-BEATS blocks was selected from {1,…, 10}, the number of layers was selected from {1, 2, 3, 4, 5}, whereas the dropout rate was tuned with values from {0, 0.1, 0.2, 0.3, 0.4 0.5}. The tuned values are shown in Table 9 for the LSTM models and Table 10 for the N-BEATS models. Overall, the LSTM-based models worked best with only 2 or 3 fully connected layers in all scenarios, whereas the N-BEATS-based models worked best with 4 or 5 fully connected layers. The tuned number of blocks in the N-BEATS-based models varied between 3 and 5, depending on the scenario and the unrestricted vs. inertial case. The tuned dropout rates varied a lot between scenarios for the LSTM-based models, with rates ranging from 0 to 0.5, whereas the tuned rates for N-BEATS-based models varied between 0.2 and 0.5.

The size of the LSTM state was tuned to 32, whereas the size of each fully connected layer was tuned to 64, which is substantially smaller than the hidden size of 512 used in the original N-BEATS model [9]. For the carbohydrates without bolus scenario Carbs${}^{\left(-b\right)}$, due to the much smaller number of examples, we reduced the number of units in the LSTM networks and fully connected layers by a factor of 2. The same hyper-parameters that were tuned on the general carbohydrate recommendation scenario Carbs${}^{\left(\pm b\right)}$ were used for Carbs${}^{\left(-b\right)}$.

The largest models discussed in this paper are the N-BEATS-based inertial models for the two bolus recommendation scenarios. With close to 11,000 parameters, these models take up less than 1 MB of memory, which indicates that they could be deployed on a wearable device or smartphone. The tuning, training, and testing experiments were run on an architecture with a 12-core Intel i7 CPU and 64 GB of RAM, supported by an NVIDIA GeForce GTX 1080TI GPU with 11 GB of memory. Training a single N-BEATS-based model on this architecture takes approximately 40 s, whereas making one recommendation prediction at test time takes only 0.1 s. Pre-training the model on the training data from all 12 subjects takes approximately 8 min, while the hyper-parameter tuning procedure, which requires training multiple times, takes approximately 30 h for each recommendation scenario.

## 7. Experimental Results

Table 11 shows the results for the two baselines and the two neural architectures: the LSTM-based (Figure 1 and Figure 2) and the N-BEATS-based (Figure 3). Across all scenarios and for both example classes, the neural models outperform both baselines, often by a wide margin. Furthermore, the N-BEATS-based models outperform their LSTM-based counterparts across all evaluations with inertial examples, which are the ones with the most practical utility. In general, there is little difference between the best model scores and the average model scores, which means that the model performance is relatively stable with respect to the random initialization of the network parameters.

For the prediction of carbohydrates without an associated bolus scenario Carbs${}^{\left(-b\right)}$, the improvement brought by the two neural models over the two baselines was less substantial, which is to be expected for two reasons. First, the baselines do much better in this scenario than in the more general carbohydrate recommendation scenario Carbs${}^{\left(\pm b\right)}$ because most of the carb intakes are relatively small, e.g., hypo correction events where subjects are advised to eat a fixed amount of carbohydrate. Second, and most importantly, the number of training carbohydrate events and their associated examples in the Carbs${}^{\left(-b\right)}$ scenario is much smaller than in the Carbs${}^{\left(\pm b\right)}$ scenario (Table 1), which makes ML models much less effective.

Figure 4 shows boxplots of the absolute error of the N-BEATS model for each of the four recommendation scenarios per subject. Overall, the median error is lower than the average error, which is skewed by outliers. Error analysis reveals that outliers are largely caused by noise in the data reported by subjects in the smartphone interface, or in some cases in the bolus wizard. For example, in the Carbs${}^{\left(\pm b\right)}$ scenario, where the median errors are generally small, subject 540 has an average error that is significantly higher than the median due to 13 particularly large outliers. These outliers have an average absolute error of about 64 g, which were clipped to 50 carbs in the boxplot. All these 13 examples stem from the same meal event of 90 g that the subject reported for 11am. However, the meal timestamp appears to be wrong, because the bolus of 8.5 units associated with the meal had a timestamp nearly an hour and a half after the meal. The pre-processing procedure was not able to correct the meal timestamp because it only looks for meals and boluses that are at most one hour apart. Therefore, due to the meal being reported much earlier than when it really happened, the subsequent BGL did not change much and so the N-BEATS model output a small number of carbs, resulting in a large absolute error. Subject 596 has the largest box (representing the range in which 50% of the absolute errors fall). This is largely because the pre-processing procedure could not be applied to subject 596’s data, since the subject did not use the bolus wizard. For the Carbs${}^{\left(-b\right)}$ scenario, the boxes are generally larger than those in the Carbs${}^{\left(\pm b\right)}$ plot, which is consistent with the results in Table 11. For the Bolus${}^{\left(\pm c\right)}$ scenario, the boxplot for subject 575 is not shown because this subject has a very small number of test examples, making the boxplot flat and uninformative. In both Bolus${}^{\left(\pm c\right)}$ and Bolus${}^{\left(+c\right)}$ scenarios, subject 563 had several large outliers, for which the average absolute error is 9.7 units and 6.5 units, respectively. This subject had a few boluses that had non-zero carbohydrate input to the BW, but had no carbohydrate events temporally close to the bolus. As such, the pre-processing procedure added a meal event 10 min after the bolus containing the number of carbs that were entered into the bolus wizard. Based on the BGL behavior, it is likely that the subject either did not eat in these situations, or that the number of carbs in the meal that the subject actually ate differed significantly from the amount of carbs they entered into the BW. Overall, subjects 563 and 584 have the largest errors in the Bolus${}^{\left(\pm c\right)}$ scenario. According to the statistics reported in Table 2, of all subjects used at test time, these 2 subjects have the highest average amount of insulin per bolus, at 8.0 and 7.3 units, respectively. Subject 567 is not included in this analysis because they had 0 carbohydrate events in their test data (as explained in Section 6.1).

In all experiments reported so far, one model was trained for all prediction horizons, using the value of τ∈{30,35,…,90} as an additional input feature. This global model was then tested on examples from all prediction horizons. To determine if transfer learning happens among different prediction horizons, for each value of τ∈{30,45,60,75,90} at test time, we compare the performance of the globally trained model vs. the performance of a model trained only on examples for that particular prediction horizon, using inertial examples for both. We chose the inertial case for this experiment because it corresponds better to the intended use of a carbohydrate or bolus recommendation system. Furthermore, we experiment only with the N-BEATS-based model because of its better performance in the inertial case. The results in Table 12 show transfer learning clearly happening for the carbohydrate recommendation Carbs${}^{\left(\pm b\right)}$ and bolus given carbs recommendation Bolus${}^{\left(+c\right)}$ scenarios, where the models trained on all prediction horizons outperform those trained only on a specific prediction horizon when evaluated on that prediction horizon. For the bolus recommendation scenario Bolus${}^{\left(-c\right)}$ (i.e., Bolus${}^{\left(\pm c\right)}$ inertial) the results were mixed, with transfer learning being clear only for the short τ=30 time horizon. Transfer learning results for the Carbs${}^{\left(-b\right)}$ scenario are not calculated due to the lack of a sufficient number of training examples for each prediction horizon. The results in Table 12 also show that in general there is a downward trend in the error as the time horizon is increased, which makes the evaluation on longer prediction horizons worth pursuing in future work.

## 8. Conclusions and Future Directions

This paper presents a general LSTM-based neural architecture, composed of two chained LSTMs and a fully connected network, for training models that make recommendations to aid in the self-management of type 1 diabetes. It focuses on recommendations for carbohydrate to consume or insulin to take to achieve a target blood glucose level, but recommendations might also be made for any type of quantitative event impacting BGLs. A deep residual N-BEATS-based architecture was also developed, using the chained LSTMs as a component in its block structure. Experimental evaluations show that the proposed neural architectures substantially outperform a global average baseline as well as a time-of-day-dependent baseline, with the N-BEATS-based models outperforming the LSTM-based counterparts in all evaluations with inertial examples. The trained models are shown to benefit from transfer learning and from a pre-processing of meal events that anchors their timestamps shortly after their corresponding boluses. Overall, these results suggest that the proposed recommendation approaches hold significant promise for easing the complexity of self-managing blood glucose levels in type 1 diabetes. Potential future research directions include investigating the proposed pre-processing of carbohydrate events for blood glucose level prediction and exploring the utility of the two neural architectures for recommending exercise. To enable reproducibility and future experimental comparions, we are making our code publicly available on the SmartHealth Lab web site at http://smarthealth.cs.ohio.edu/nih.html.

## Figures and Tables

**Figure 1 sensors-21-03303-f001:**
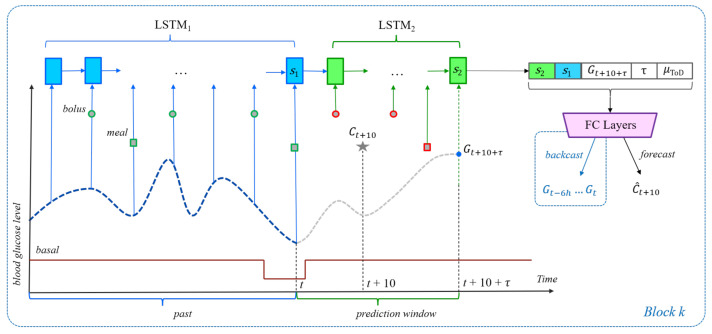
The neural network architecture for the carbohydrate recommendation scenario. The dashed blue line plots BG levels, while the solid red line represents the basal rate of insulin. The gray star represents the meal event at time t+10. Other meals are represented by squares, whereas boluses are represented by circles. Meals and boluses with a red outline can appear only in unrestricted examples. The blue ${\mathrm{LSTM}}_{1}$ units receive input from time steps in the past. The green ${\mathrm{LSTM}}_{2}$ units receive input from time steps in the prediction window. The purple block stands for the fully connected layers of the FCN that computes the prediction.

**Figure 2 sensors-21-03303-f002:**
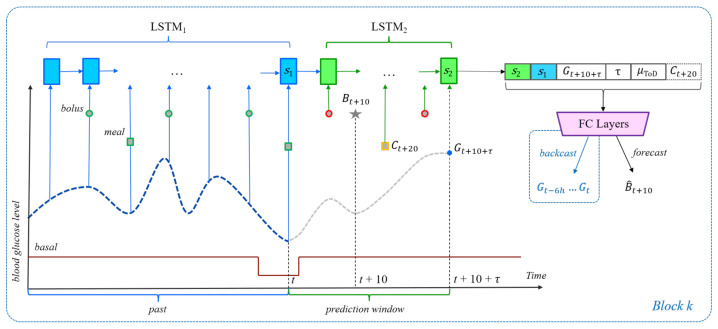
The general neural network architecture for the bolus and bolus given carbs recommendation scenarios. The architecture itself is similar to that shown in Figure 1. The gray star now represents the bolus at time t+10. For the bolus recommendation scenario, the events outlined in red or orange are not allowed in inertial examples. However, in the bolus given carbs scenario, the meal event $C_{t+20}$ shown with the yellow outline is an important part of each example, be it inertial or unrestricted. As such, in this scenario, the dashed $C_{t+20}$ becomes part of the input to the FCN.

**Figure 3 sensors-21-03303-f003:**
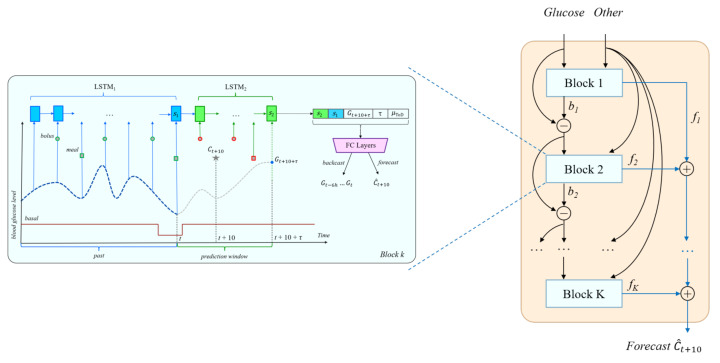
The N-BEATS inspired deep residual architecture for carbohydrate recommendation. A similar architecture is used for bolus and bolus given carbs recommendations.

**Figure 4 sensors-21-03303-f004:**
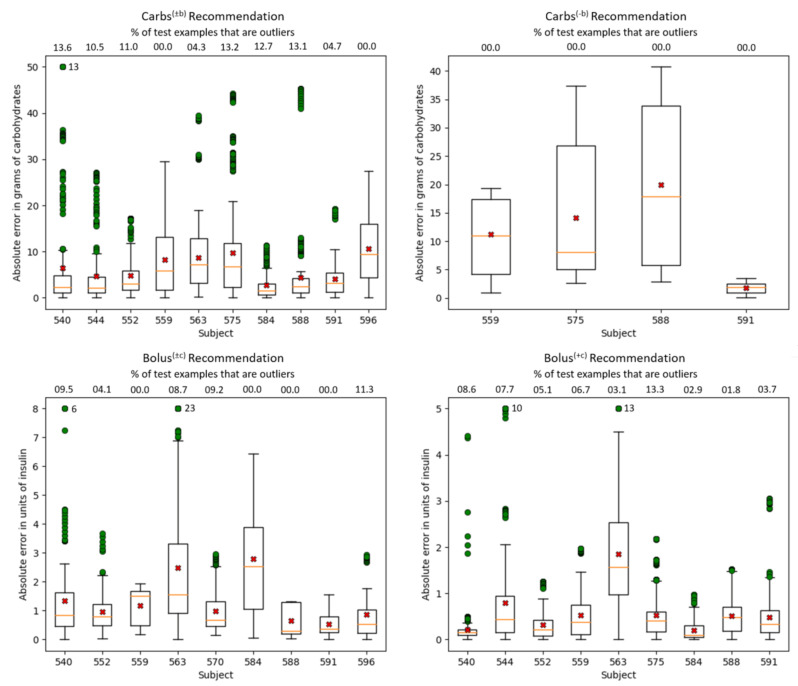
Boxplots showing the absolute error per subject for each recommendation scenario achieved by the N-BEATS.best model in the inertial scenario. The orange lines within each box represent the median absolute errors, while the red crosses represent the average absolute errors. The green circles represent outliers. To avoid stretching the figures, outliers were clipped at 25 for the Carbs${}^{\left(\pm b\right)}$ scenario, 8 for the Bolus${}^{\left(\pm c\right)}$ scenario, and 5 for the Bolus${}^{\left(+c\right)}$ scenario. The number of clipped outliers is shown next to the subject’s largest outlier. Above the top line is shown, for each subject, the percentage of test examples that are outliers.

**Table 1 sensors-21-03303-t001:** Per subject and total meal and carbohydrate per meal statistics: Minimum, Maximum, Median, Average, and Standard Deviation (StdDev). Carbs${}^{\left(\pm b\right)}$ refers to all carbohydrate intake events; Carbs${}^{\left(-b\right)}$ refers to carbohydrate intakes without a bolus. Statistics are shown for the 2018 subset, the 2020 subset, and for the entire OhioT1DM dataset.

			Carbs Per Meal (g)				
**Subject**	**Carbs** ${}^{\left(\pm b\right)}$	**Carbs** ${}^{\left(-b\right)}$	**Minimum**	**Maximum**	**Median**	**Average**	**StdDev**
559	215	83	8.0	75.0	30.0	35.5	15.5
563	225	28	5.0	84.0	31.0	33.8	18.0
570	174	39	5.0	200.0	115.0	106.1	41.5
575	297	122	1.0	110.0	40.0	40.0	22.0
588	268	73	2.0	60.0	20.0	22.7	14.6
591	264	60	3.0	77.0	28.0	31.5	14.1
2018 Total	1443	405	1.0	200.0	33.0	41.5	32.7
540	234	14	1.0	110.0	40.0	50.2	29.8
544	206	41	1.0	175.0	60.0	68.7	36.3
552	271	25	3.0	135.0	26.0	36.7	29.3
567	207	5	20.0	140.0	67.0	67.0	21.5
584	233	44	15.0	78.0	60.0	54.6	11.6
596	300	277	1.0	64.0	25.0	25.1	14.0
2020 Total	1451	406	1.0	175.0	42.0	48.2	29.5
Combined Total	2894	811	1.0	200.0	39.0	44.9	31.3

**Table 2 sensors-21-03303-t002:** Per subject and total boluses and insulin units statistics: Minimum, Maximum, Median, Average, and Standard Deviation (StdDev). Bolus${}^{\left(\pm c\right)}$ refers to all bolus events; Bolus${}^{\left(+c\right)}$ refers to bolus events associated with a meal. Statistics are shown for the 2018 subset, the 2020 subset, and for the entire OhioT1DM dataset.

			Insulin Per Bolus (u)				
**Subject**	**Bolus** ${}^{\left(\pm c\right)}$	**Bolus** ${}^{\left(+c\right)}$	**Minimum**	**Maximum**	**Median**	**Average**	**StdDev**
559	186	132	0.1	9.3	3.6	3.7	1.9
563	424	197	0.1	24.7	7.8	8.0	4.2
570	1345	132	0.2	12.1	1.3	1.8	2.1
575	271	175	0.1	12.8	4.4	4.1	3.0
588	221	195	0.4	10.0	3.5	4.3	2.3
591	331	204	0.1	9.4	2.9	3.1	1.8
2018 Total	2758	1035	0.1	24.7	1.9	3.5	3.4
540	521	220	0.1	11.4	2.0	3.0	2.8
544	264	149	0.7	22.5	5.0	6.5	4.9
552	426	246	0.1	16.0	2.8	3.9	3.3
567	366	202	0.2	25.0	11.4	12.0	5.8
584	311	188	0.1	16.2	9.1	7.3	3.1
596	230	0	0.2	7.6	3.3	3.0	1.5
2020 Total	2118	1169	0.1	25.0	4.0	5.8	5.0
Combined Total	4876	2204	0.1	25.0	2.9	4.5	4.3

**Table 3 sensors-21-03303-t003:** *Inertial* (*I*) examples by recommendation scenario and prediction horizon. Carbs${}^{\left(\pm b\right)}$ refers to all carbohydrate intake events; Carbs${}^{\left(-b\right)}$ refers to carbohydrate intakes without a bolus.

	**Carbs** ${}^{\left(\pm b\right)}$ **Recommendation**				**Carbs** ${}^{\left(-b\right)}$ **Recommendation**			
**Horizon**	**Training**	**Validation**	**Testing**	**Total** ***I***	**Training**	**Validation**	**Testing**	**Total** ***I***
τ=30	1192	340	331	1863	265	53	40	358
τ=45	1156	334	321	1811	255	51	40	346
τ=60	1121	318	315	1754	243	50	40	333
τ=75	1057	301	293	1651	226	44	34	304
τ=90	975	279	278	1532	200	40	31	271
All 13 horizons	14,343	4103	4007	22,453	3100	620	486	4206
	**Bolus${}^{\left(\pm c\right)}$ Recommendation**				**Bolus${}^{\left(+c\right)}$ Recommendation**			
**Horizon**	**Training**	**Validation**	**Testing**	**Total** ***I***	**Training**	**Validation**	**Testing**	**Total** ***I***
τ=30	461	160	143	764	856	267	271	1394
τ=45	416	142	124	682	833	259	258	1350
τ=60	368	124	104	596	816	253	249	1318
τ=75	303	102	96	501	790	243	243	1276
τ=90	271	90	86	447	743	234	229	1206
All 13 horizons	4732	1606	1423	7761	10,514	3269	3249	17,032

**Table 4 sensors-21-03303-t004:** *Unrestricted* (U) examples by recommendation scenario, also showing, in the last column, the total number of non-inertial (U−I) examples. Carbs${}^{\left(\pm b\right)}$ refers to all carbohydrate intake events; Carbs${}^{\left(-b\right)}$ refers to carbohydrate intakes without a bolus.

Scenario	Training	Validation	Testing	Total *U*	Total U−I
Carbs${}^{\left(\pm b\right)}$	17,937	5106	4943	27,986	5533
Carbs${}^{\left(-b\right)}$	4140	853	624	5617	1411
Bolus${}^{\left(\pm c\right)}$	19,640	6279	6136	32,055	24,294
Bolus${}^{\left(+c\right)}$	12,052	3784	3816	19,652	2620

**Table 5 sensors-21-03303-t005:** Results with pre-processing of meals (pre) vs. original raw data for meal events (raw), for the carbohydrate recommendation scenario Carbs${}^{\left(\pm b\right)}$ on unrestricted examples. pre${}^{+}$ refers to using all pre-processed meals (shifted original meals and added meals), whereas pre${}^{-}$ does not use meals added by the pre-processing procedure. We show results for different combinations of pre-processing options during Training and evaluation on Validation data, e.g., the first row indicates raw data were used during both training and evaluation. The symbol ${}^{†}$ indicates a *p*-value < 0.03 when using a one-tailed *t*-test to compare against the results without pre-processing (raw).

	Pre-Processing			
	Training	Validation	RMSE	MAE
N-BEATS.mean	raw	raw	13.42	10.32
	pre${}^{+}$	pre${}^{-}$	9.38 ${}^{†}$	6.59 ${}^{†}$
	pre${}^{+}$	pre${}^{+}$	8.84 ${}^{†}$	6.16 ${}^{†}$
N-BEATS.best	raw	raw	12.32	9.28
	pre${}^{+}$	pre${}^{-}$	8.48 ${}^{†}$	5.90 ${}^{†}$
	pre${}^{+}$	pre${}^{+}$	8.12 ${}^{†}$	5.53 ${}^{†}$

**Table 6 sensors-21-03303-t006:** Results with pre-processing of meals (pre) vs. original raw data for meal events (raw), for the Bolus${}^{\left(\pm c\right)}$ recommendation scenario on unrestricted examples. All meals (shifted or added) are used for the pre-processed data. The symbol ${}^{†}$ indicates a *p*-value < 0.01 when using a one-tailed *t*-test to compare against the results without pre-processing (raw).

	Pre-Processing	RMSE	MAE
N-BEATS.mean	raw	1.85	1.41
	pre	1.30 ${}^{†}$	0.92 ${}^{†}$
N-BEATS.best	raw	1.81	1.32
	pre	1.22 ${}^{†}$	0.84 ${}^{†}$

**Table 7 sensors-21-03303-t007:** Performance of the LSTM- and N-BEATS-based models, with (+) and without (−) the final state $s_{1}$ of LSTM${}_{1}$ as part of the input to the FC Layers.

LSTM.mean		RMSE	MAE		N-BEATS.mean		RMSE	MAE
Carbs${}^{\left(\pm b\right)}$	$-s_{1}$	10.14	7.56		Carbs${}^{\left(\pm b\right)}$	$-s_{1}$	10.27	7.58
	$+s_{1}$	8.99	6.57			$+s_{1}$	8.84	6.16
Bolus${}^{\left(\pm c\right)}$	$-s_{1}$	1.33	0.97		Bolus${}^{\left(\pm c\right)}$	$-s_{1}$	1.33	0.85
	$+s_{1}$	1.41	1.03			$+s_{1}$	1.30	0.92

**Table 8 sensors-21-03303-t008:** N-BEATS-based model results, with a *separate* vs. *joint* final fully connected layer for computing backcast and forecast values.

N-BEATS.mean		RMSE	MAE
Carbs${}^{\left(\pm b\right)}$	*separate*	8.77	6.48
	*joint*	8.84	6.16
Bolus${}^{\left(\pm c\right)}$	*separate*	1.32	0.94
	*joint*	1.30	0.92

**Table 9 sensors-21-03303-t009:** Tuned hyper-parameters for the LSTM-based models.

		Hyper-Parameters	
**Scenario**	**Examples**	**FC Layers**	**Dropout**
Carbs${}^{\left(\pm b\right)}$	Inertial	3	0.1
	Unrestricted	3	0.1
Bolus${}^{\left(\pm c\right)}$	Inertial	3	0.0
	Unrestricted	2	0.3
Bolus${}^{\left(+c\right)}$	Inertial	2	0.2
	Unrestricted	2	0.5

**Table 10 sensors-21-03303-t010:** Tuned hyper-parameters for the N-BEATS-based models.

		Hyper-Parameters		
**Scenario**	**Examples**	**Blocks**	**FC Layers**	**Dropout**
Carbs${}^{\left(\pm b\right)}$	Inertial	5	2	0.3
	Unrestricted	3	3	0.3
Bolus${}^{\left(\pm c\right)}$	Inertial	5	4	0.2
	Unrestricted	4	4	0.2
Bolus${}^{\left(+c\right)}$	Inertial	5	4	0.5
	Unrestricted	3	5	0.2

**Table 11 sensors-21-03303-t011:** Results for each recommendation scenario, for both classes of examples. The simple † indicates a *p*-value < 0.05 when using a one-tailed *t*-test to compare against the baseline results; the double ${}^{‡}$ indicates statistical significance for comparison against the baselines as well as against the competing neural method; the ${}^{↑}$ indicates significant with respect to the Global Average baseline only.

	**Inertial**		**Unrestricted**	
**Carbs** ${}^{\left(\pm b\right)}$ **Recommendation**	**RMSE**	**MAE**	**RMSE**	**MAE**
Global Average	20.90	17.30	20.68	17.10
ToD Average	20.01	15.78	19.82	15.68
LSTM.mean	11.55	7.81	10.99	7.40
LSTM.best	10.95	7.50	10.50	7.31
N-BEATS.mean	9.79 ${}^{‡}$	6.45 ${}^{‡}$	10.34	7.04
N-BEATS.best	9.92	6.56	10.07 ${}^{†}$	6.75 ${}^{†}$
	**Inertial**		**Unrestricted**	
**Carbs** ${}^{\left(\pm b\right)}$ **Recommendation**	**RMSE**	**MAE**	**RMSE**	**MAE**
Global Average	15.92	13.71	14.66	12.19
ToD Average	15.55	13.45	14.27	11.93
LSTM.mean	14.02	11.47	14.70	12.27
LSTM.best	13.75	10.92	14.94	12.57
N-BEATS.mean	13.76	11.42	13.69 ${}^{↑}$	11.09 ${}^{↑}$
N-BEATS.best	14.52	11.78	14.17	11.47
	**Inertial**		**Unrestricted**	
**Bolus** ${}^{\left(\pm c\right)}$ **Recommendation**	**RMSE**	**MAE**	**RMSE**	**MAE**
Global Average	2.40	2.13	2.84	2.30
ToD Average	2.21	1.86	2.71	2.17
LSTM.mean	1.75	1.35	1.53	1.10
LSTM.best	1.70	1.30	1.50	1.05
N-BEATS.mean	1.56 ${}^{†}$	1.20 ${}^{‡}$	1.49 ${}^{†}$	1.04
N-BEATS.best	1.65	1.26	1.51	1.03 ${}^{†}$
	**Inertial**		**Unrestricted**	
**Bolus** ${}^{\left(+c\right)}$ **Recommendation**	**RMSE**	**MAE**	**RMSE**	**MAE**
Global Average	3.00	2.35	3.04	2.39
ToD Average	2.87	2.21	2.90	2.25
LSTM.mean	1.02	0.73	1.00	0.73
LSTM.best	0.94	0.67	1.00 ${}^{†}$	0.72 ${}^{†}$
N-BEATS.mean	0.89	0.65	1.11	0.82
N-BEATS.best	0.85 ${}^{†}$	0.61 ${}^{†}$	1.06	0.78

**Table 12 sensors-21-03303-t012:** Comparison between models trained on all prediction horizons vs. one prediction horizon τ, when evaluated on the prediction horizon τ. The symbol ${}^{†}$ indicates a *p*-value < 0.05 when using a one-tailed *t*-test to compare against the one prediction horizon results.

		**Carbs** ${}^{\left(\pm b\right)}$ **Recommendation**											
		τ=30		τ=45		τ=60		τ=75		τ=90		**Average**	
	**Trained on**	**RMSE MAE**		**RMSE MAE**		**RMSE MAE**		**RMSE MAE**		**RMSE MAE**		**RMSE MAE**	
N-BEATS.mean	One τ	9.74	6.72	10.24	6.89	10.06	6.85	10.52	7.19	9.82	6.73	10.08	6.88
	All τ	9.96	6.57	9.98	6.56	9.84	6.50	9.55 ${}^{†}$	6.30 ${}^{†}$	9.37	6.22	9.74	6.43
N-BEATS.best	One τ	9.92	6.70	10.39	6.90	10.21	6.88	10.62	7.18	9.92	6.66	10.21	6.86
	All τ	9.84	6.50	9.94	6.56	10.02	6.57	9.76	6.34 ${}^{†}$	9.43	6.08	9.80	6.41
		**Bolus** ${}^{\left(-c\right)}$ **Recommendation**											
		τ=30		τ=45		τ=60		τ=75		τ=90		**Average**	
	**Trained on**	**RMSE MAE**		**RMSE MAE**		**RMSE MAE**		**RMSE MAE**		**RMSE MAE**		**RMSE MAE**	
N-BEATS.mean	One τ	1.82	1.42	1.57	1.24	1.51	1.24	1.37	1.10	1.40	1.17	1.53	1.23
	All τ	1.75	1.33	1.61	1.24	1.47	1.17 ${}^{†}$	1.38	1.10	1.28	1.03 ${}^{†}$	1.50	1.17 ${}^{†}$
N-BEATS.best	One τ	1.77	1.37	1.54	1.21	1.51	1.23	1.38	1.10	1.34	1.11	1.51	1.20
	All τ	1.72	1.28	1.75	1.33	1.58	1.23	1.45	1.12	1.44	1.13	1.59	1.22
		**Bolus** ${}^{\left(+c\right)}$ **Recommendation**											
		τ=30		τ=45		τ=60		τ=75		τ=90		**Average**	
	**Trained on**	**RMSE MAE**		**RMSE MAE**		**RMSE MAE**		**RMSE MAE**		**RMSE MAE**		**RMSE MAE**	
N-BEATS.mean	One τ	0.98	0.73	0.91	0.69	0.91	0.69	0.95	0.74	0.93	0.72	0.94	0.71
	All τ	0.95	0.68	0.87	0.65	0.86	0.65	0.87 ${}^{†}$	0.65 ${}^{†}$	0.86 ${}^{†}$	0.64 ${}^{†}$	0.88 ${}^{†}$	0.65 ${}^{†}$
N-BEATS.best	One τ	0.94	0.69	0.91	0.69	0.92	0.68	0.93	0.71	0.91	0.70	0.92	0.69
	All τ	0.94	0.66	0.84	0.62 ${}^{†}$	0.82 ${}^{†}$	0.59 ${}^{†}$	0.82 ${}^{†}$	0.61 ${}^{†}$	0.83 ${}^{†}$	0.61 ${}^{†}$	0.85 ${}^{†}$	0.62 ${}^{†}$

## Data Availability

The OhioT1DM dataset is available under a Data Use Agreement (DUA) at http://smarthealth.cs.ohio.edu/OhioT1DM-dataset.html.

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
