# Peer review of "LSTMs and Deep Residual Networks for Carbohydrate and Bolus Recommendations in Type 1 Diabetes Management"

_sensors, 2021, doi:10.3390/s21093303_

Round 1

Reviewer 1 Report

I have no comments. It is a very good work and it is perfectly explained.

Author Response

Thank you for the positive feedback (no additional response needed).

Reviewer 2 Report

I really enjoyed reviewing this manuscript. The authors propose a deep learning model to make recommendations about carbohydrates and insulin boluses in type 1 diabetes therapy. The novelty of the approach is clear. The paper is well written, background, methods and results are clearly explained.

I have some minor comments.

The approach presented is very promising, however I think that the max prediction horizon of 90 min would limit the applicability of the model in real life. Indeed, the post-prandial BG curve is much longer than 90 minutes, and the velocity of meal absorption strongly depends on the meal composition. Therefore, I think that it would be difficult for a patient to set a target BGL at 90 min after the meal in the scenario of meal bolus recommendations. It is also difficult to determine a target for the bolus only scenario, because typically 90 min after the bolus there is still much insulin on board. I would like the authors to comment these applicability issues in the discussion.

Can the authors clarify why in the preprocessing step the time of meal boluses is forced to be 10 min before the meal? Is this required by the specific network architecture?

It would be nice to see boxplot/histograms of RMSE and MAE values for all the examples in the test set. This would allow to appreciate if the model is making large errors for some examples, or if the performance are generally acceptable for all the examples (or the majority of them).

Can the authors also comment about the computation time required by their models? Is this a limitation for model implementation, e.g. in a mobile app?

Please check the labels in Table 3 for the scenarios with bolus recommendation. Is it correct that the scenario with Bolus+/-c has less examples than the scenario with Bolus+c?

Please provide measurement units for RMSE and MAE.

Please clarify the difference between “Train” and “Devel” in Table 5.

Definition of abbreviation “ML” is missing.

Reviewer 3 Report

The focus of the manuscript is novel and timely. Authors present work in blood glucose level prediction and automated bolus calculators at three scenarios. The methodology and results are well organized and presented and offer future research direction.

Author Response

(The authors gave the same response as above.)

Reviewer 4 Report

It is a well written manuscript. The authors present the implementation of two deep learning models for carbohydrate and insulin dosage recommendation for self-regulation of blood glucose in patients with type 1 diabetes. The chained LSTM recurrent network and the LSTM block integrated residual network which are proposed are improvements of previous existed networks, they treat however a more general class of examples (unrestricted examples) in which bolus and meal events are allowed in the prediction window.

It is a well presented work with a clear methodology for a complex problem. All the aspects of the work (dataset, proposed preprocessing, scenarios, models and results) are analytically presented and described in detail. The results while outperforming the baselines seem promising.

Author Response

(The authors gave the same response as above.)
